# SIMP Dark Matter

**Yonit Hochberg**

Racah Institute of Physics, Hebrew University of Jerusalem, Jerusalem 91904, Israel

yonit.hochberg@mail.huji.ac.il

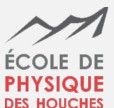

*Part of the Dark Matter*
*Session 118 of the Les Houches School, July 2021*
*published in the Les Houches Lecture Notes Series*

## Abstract

These notes summarize the zoom-course on Strongly Interacting Massive Particle (SIMP) dark matter given at the Les Houches Summer School in the summer of 2021. An alternative title for this course would be 'SIMPS, Cannibals and ELDERs', where our focus is on high-point interactions amongst dark matter particles. The spirit of the course is to give students a taste of these exciting developments in dark matter, while primarily teaching tricks and methods that are tough to captivate when reading textbooks or papers. These notes are written in similar spirit: I will put emphasis on how quantities scale and how to perform back of the envelope estimates, which should serve to help you when you develop the next great dark matter idea.

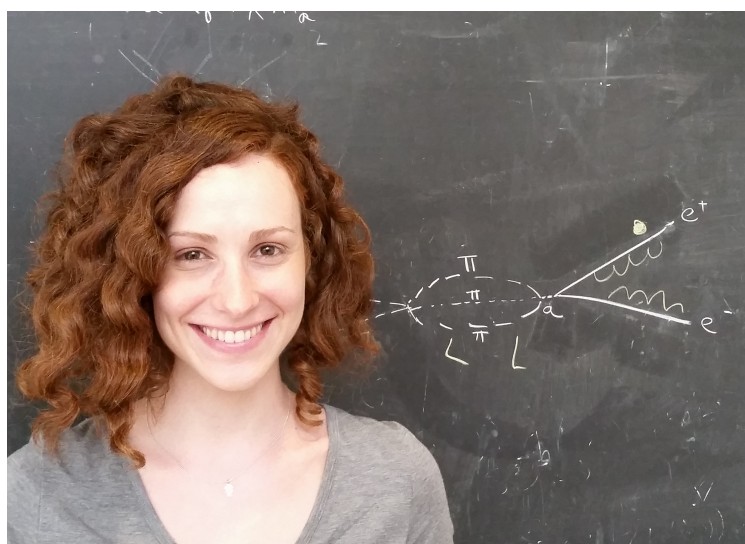

# 1 Setting the Stage

Since first being postulated in the 1930's, we have accumulating indirect evidence for the existence of dark matter (DM). What we have learned thus far is the following:

- Our universe is dark: nearly 27% of the energy content of the universe is in the form of DM. In particle physics language, DM has five times the mass density of baryons: $\rho_{\text{DM}} \simeq 5\rho_{\text{baryons}}$.

- We know that DM is massive, but we have no idea what its mass is. It can be as heavy as $\sim 10^{50}$ GeV or as light as $\sim 10^{-30}$ GeV, spanning some 80 (!) orders of magnitude of possibilities.

- DM shouldn't interact too strongly with the known forces of quantum electrodynamics (QED) and quantum chromodynamics (QCD), or else we would have detected it already.

- DM shouldn't interact too strongly with itself, as that would distort dynamics in DM halos. ($\iff$ though could this be a possible signal?)

- DM plays an important role in our cosmological history; we wouldn't be here without it!

We want to understand: What is DM? What are mechanisms in the early universe that set its relic abundance? Which models can realize these mechanisms? What are the existing constraints, and how can we detect it?

## 2  Early Universe Cheat Sheet

To develop our tool kit for answering these questions, here is a crash course on 'Early Universe 101'. (A great place for full details is *e.g.* [1]). Our universe is expanding. A length $\ell$ gets scaled to $\tilde{\ell} = a\ell$, with $a$ the scale factor (which is time $t$ dependent), and volume expands like $a^3$. We then have

$$ds^2 = dt^2 - a(t)^2 dx^2\,, \tag{1}$$

and define Hubble $H$ to be

$$H \equiv \frac{\dot{a}}{a} = \frac{1}{a}\frac{\partial a}{\partial t}\,. \tag{2}$$

From the first Friedman equation we have $H^2 = \frac{\rho}{3M_{\mathrm{Pl}}^2}$ with $M_{\mathrm{pl}}$ the reduced Planck mass, and since $\rho \propto T^4$ for blackbody radiation, we have the important scaling

$$H \sim \frac{T^2}{M_{\mathrm{Pl}}}\,. \tag{3}$$

The early universe is a thermal environment, and particles have phase space distributions which we can integrate over to obtain number density and energy density of different species. For relativistic (R) cases, when the mass is smaller than the temperature $m \ll T$, we have

$$n \sim T^3\,, \qquad \rho \sim T^4\,, \qquad \text{for relativistic particles (R)}\,, \tag{4}$$

while for non-relativistic (NR) cases $m \gg T$,

$$n \sim (mT)^{3/2}e^{-(m-\mu)/T}\,, \qquad \rho \sim mn\,, \qquad \text{for non−relativistic particles (NR)}\,. \tag{5}$$

In particular, when number-changing processes are fast, the chemical potential $\mu$ vanishes, and the number density is exponentially suppressed, $n \propto e^{-m/T}$. Finally, the entropy density $s$ is governed by the relativistic degrees of freedom (dof) and scales as

$$s \sim T^3\,. \tag{6}$$

We can now write the Boltzmann equation (BE) that governs the evolution of the system over time. If we first consider a system with no collisions (namely free particles), the total number of particles in the system experiences no change, and so $\partial N/\partial t = 0$. Writing the number of particles as density times volume $N = nV$, we then have

$$\frac{\partial(nV)}{\partial t} = V\frac{\partial n}{\partial t} + n\frac{\partial V}{\partial t} = 0\,. \tag{7}$$

Recalling that $V \propto a^3$ and that $H = \dot{a}/a$, we arrive at

$$\frac{\partial n}{\partial t} + 3nH = 0\,. \tag{8}$$

If the particles are not free and interactions are present, the right hand side contains the collision terms $C[n]$:

$$\frac{\partial n}{\partial t} + 3nH = -C[n]\,. \tag{9}$$

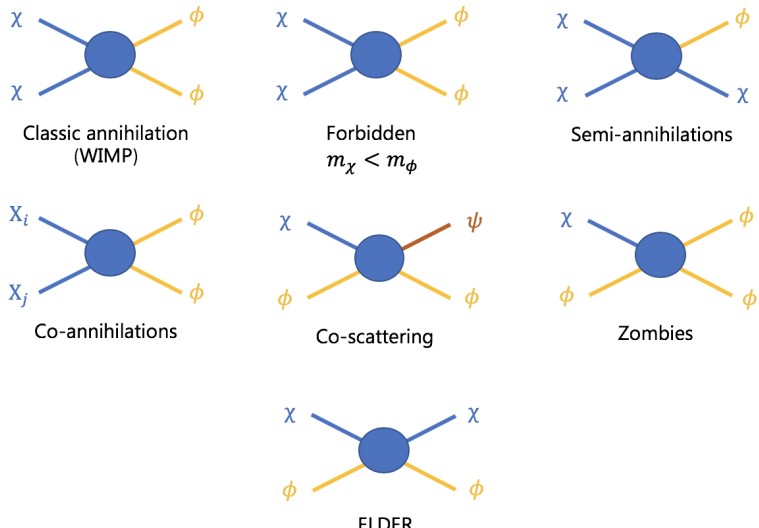

Figure 1: Some examples of the $2 \to 2$ zoo: WIMP annihilations, forbidden channels [2,3], semi-annihilations [4], co-annihilations [2], co-scattering [5], zombies [6] and ELDERs [7]. Time flows from left to right.

## 3 WIMPs

We are now ready to discuss different mechanisms for DM freezeout, namely various types of processes in the early universe that can set the relic abundance of DM. Some $2 \to 2$ processes which can do this, what we call the $2 \to 2$ zoo, are shown in Fig. 1. Throughout these notes, $\chi$ will denote the DM candidate, and time will always flow from left to right in diagrams. We begin by working through the well-known WIMP case, to phrase it in a way that will be very useful as we move beyond the WIMP.

It is no secret that the star of the show for the last $\sim 40$ years has been the WIMP. Here, two DM particles annihilate into two other particles, $\chi\chi \to \phi\phi$. If this processes is fast, it sets the chemical potentials to be equal, and since we will assume $\phi$ has zero chemical potential, $\mu_\chi = \mu_\phi = 0$. We further assume $\phi$ shares a temperature with the bath. What is the collision term for the BE of this system? Roughly, the rate for annihilation is equal to the thermally average cross section times the number density (more accurately, the flux) of the particle you have to meet, and we take into account both the forward annihilation process as well as the backwards process of production:

$$\frac{\partial n_\chi}{\partial t} + 3n_\chi H = -\langle\sigma v\rangle \left(n_\chi^2 - (n_\chi^{\mathrm{eq}})^2\right), \tag{10}$$

where $\langle\sigma v\rangle \equiv \langle\sigma v\rangle_{\chi\chi \to \phi\phi}$ for the annihilation process. The first term on the RHS is the forward annihilation process, while the second term is the backreaction. In writing it this way, we have used the extremely useful trick of detailed balance: The statement that in thermal equilibrium, forward and backward processes are both rapid and should cancel out.

What happens? Instead of thinking of particle densities in a box, let's think about particle densities in a box that's expanding with the universe. The resulting evolution of this yield $Y = n/s \sim na^3$ over time, or conveniently as a

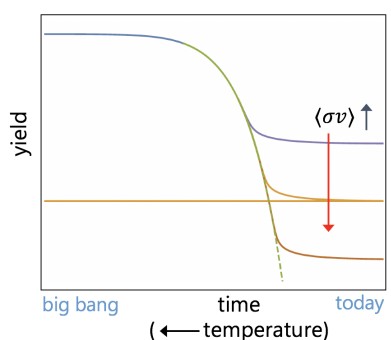

function of $x \equiv m/T$, is shown here. At early times, both processes are rapidly occurring, and the yield is constant. As the temperature drops beneath the mass of the particle, the backward production processes becomes suppressed, and DM is rapidly annihilating away. As the universe continues to cool and expand, the particles are becoming more and more dilute, until at some point the DM particles are so dilute that they no longer find each other, the annihilation processes can no longer occur, and the DM abundance is 'frozen out', leaving us with a constant amount of DM. How much DM is left is of course a function of the strength of the interaction, namely of the thermally averaged cross section. This is the standard picture for DM freeze out.

To understand when this happens, let's perform a back-of-the-envelope estimate. Freeze-out roughly occurs when the rate for the $2 \to 2$ annihilation $\Gamma_{2\to2}$ is of order Hubble $H$. The rate of annihilation has two factors in it – it is proportional to the number density of the particle the DM has to meet, and to the strength of the interaction, and this should be comparable to the expansion of the universe

$$\Gamma_{2\to2} = n_\chi \langle \sigma v \rangle \sim H \sim \frac{T^2}{M_{\text{Pl}}} \,. \tag{11}$$

To obtain understanding at the mechanism level, we parameterize the $2 \to 2$ process by

$$\langle \sigma v \rangle \equiv \frac{\alpha_{\text{eff}}^2}{m_\chi^2} \,. \tag{12}$$

The trick we will use is to relate $n_\chi$ to measured quantities. In this way we will incorporate in our estimate automatically the fact that we are interested in accounting for the observed amount of DM in our universe today. To do this, we will play with redshifts. We will use entropy conservation, which means that $S \equiv sa^3$ is constant $\Rightarrow s \sim 1/a^3 \sim T^3$. We will choose to redshift to the time of matter-radiation equality, $T_{\text{eq}} \sim 0.8$ eV. At matter-radiation equality, the energy density in photons equals that in DM and baryons combined,

$$\rho_{\text{matter}}^{\text{eq}} = \rho_\chi^{\text{eq}} + \rho_{\text{baryons}}^{\text{eq}} = \rho_\gamma^{\text{eq}} \,. \tag{13}$$

We will use the fact that $\rho_\chi \simeq 5\rho_{\text{baryons}}$ and drop the $\mathcal{O}(1)$ factors from here on to use $\rho_\chi^{\text{eq}} \simeq \rho_{\text{baryons}}^{\text{eq}}$ at equality. Denoting quantities at freezeout by subscript 'F', we then can redshift from freezeout to equality, assuming the DM is NR:

$$n_\chi(T_F) \sim n_\chi(T_{\text{eq}}) \left( \frac{T_F}{T_{\text{eq}}} \right)^3 \sim \frac{\rho_\chi(T_{\text{eq}})}{m_\chi} \frac{T_F^3}{T_{\text{eq}}^3} \sim \frac{\rho_\gamma(T_{\text{eq}})}{m_\chi} \frac{T_F^3}{T_{\text{eq}}^3} \sim \frac{T_F^3 T_{\text{eq}}}{m_\chi} \,, \tag{14}$$

where we have used that $\rho_\gamma \sim T^4$. Moving to $x_F = m_\chi/T_F$, we arrive at the very useful relationship:

$$\boxed{n_\chi(T_F) \sim \frac{T_{\text{eq}} m_\chi^2}{x_F^3} \,.} \tag{15}$$

We can now insert this along with the parametrization Eq. (12) into our freezeout condition Eq. (11):

$$\Gamma_{2\to2} = n_\chi(T_F) \cdot \frac{\alpha_{\text{eff}}^2}{m_\chi^2} \sim \frac{\alpha_{\text{eff}}^2 T_{\text{eq}}}{x_F^3} \sim H_F \sim \frac{T_F^2}{M_{\text{Pl}}} \sim \frac{m_\chi^2}{x_F^2 M_{\text{Pl}}} \,. \tag{16}$$

Solving for $m_\chi$, assuming $x_F \sim 20$, we arrive at

$$m_\chi \sim \alpha_{\text{eff}} \sqrt{T_{\text{eq}} M_{\text{Pl}}} \sim \alpha_{\text{eff}} \times (30 \text{ TeV}). \tag{17}$$

This is the mass-coupling relationship in order for the $2 \rightarrow 2$ process we studied to explain the observed relic abundance. It happens to be the case that if we plug in a coupling of order the electroweak coupling, $\alpha_{\text{eff}} \sim 10^{-2}$, then the weak scale emerges. Smaller coupling results in smaller masses; larger coupling give larger masses. Importantly, this is a complete coincidence of scales! We have two unrelated scales, $T_{\text{eq}}$ and $M_{\text{Pl}}$, which contrive together to give a geometric mean, which *if* we plug in a weak coupling, gives us the weak scale. This is the famous Weakly Interacting Massive Particle (WIMP), or in short, the WIMP miracle, which has been guiding the community for nearly four decades.

An alternative way to state this is to write Eq. (17) as

$$\langle \sigma v \rangle = \frac{\alpha_{\text{eff}}^2}{m_\chi^2} \sim \frac{1}{T_{\text{eq}} M_{\text{Pl}}}. \tag{18}$$

(See also Josh Ruderman's lecture notes from this school.)

# 4 SIMPs

In the process we just considered, what was important was how DM interacted with other particles. What if what is most important is how DM talks to itself? In this case we will find a very different kind of DM candidate [8].

Imagine that DM is the lightest state in a nearly secluded sector. We are interested in asking whether the self-interactions of DM can set its abundance. The first process one might think of is a $2 \rightarrow 2$ self-scattering process, except that won't change the number density of $\chi$ particles. The first process that can change the number of $\chi$ particles is a $3 \rightarrow 2$ process, where 3 DM particles annihilate into 2 DM particles, $\chi \chi \chi \rightarrow \chi \chi$. (Note that if 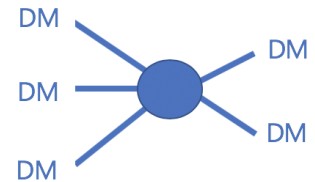 such a $3 \rightarrow 2$ processes exists between $\chi$ particles, DM must be a boson rather than a fermion.)

If such a $3 \rightarrow 2$ process is responsible for the DM abundance we observe in our universe today, what kind of masses and couplings would it point to? We can write down the BE for the system,

$$\partial_t n + 3Hn = -\langle \sigma v^2 \rangle_{3 \rightarrow 2} \left( n_\chi^3 - n_\chi^2 n_\chi^{\text{eq}} \right), \tag{19}$$

where we have taken into account both the forward $3 \rightarrow 2$ process and the backreaction of $2 \rightarrow 3$ and used detailed balance (see shortly more details). Here we have parameterized the collision term by the thermally averaged $3 \rightarrow 2$ cross section, $\langle \sigma v^2 \rangle_{3 \rightarrow 2}$. One can of course solve this equation in full, but to gain understanding, here we will use our estimating tools.

Freezeout roughly happens when the rate for the $3 \rightarrow 2$ annihilation is of order Hubble,

$$\Gamma_{3 \rightarrow 2} \sim H. \tag{20}$$

The $3 \rightarrow 2$ rate takes into account two factors,

$$\Gamma_{3 \rightarrow 2} \sim n_\chi^2 \langle \sigma v^2 \rangle_{3 \rightarrow 2}. \tag{21}$$

The first factor accounts for the fact that in order for the $3 \rightarrow 2$ process to occur, the DM particles must meet another *two* DM particles. The second factor takes into account the strength of the interaction, with the $v^2$ notation inside a reminder that the collision term cares about the *flux* of the DM particles you need to meet, which would be $(nv)^2$ with $v$ the DM velocity. Of course, this is only notation; what really enters the BE is the collision term, call its parts as you wish.

We parameterize the $3 \rightarrow 2$ cross section at the mechanism level as

$$\langle \sigma v^2 \rangle_{3 \rightarrow 2} \equiv \frac{\alpha_{\text{eff}}^3}{m_\chi^5} \,, \tag{22}$$

which carries the correct mass dimensions (see below), and use the number density of DM at freezeout from our redshift trick Eq. (15), which already accounts for the observed DM relic abundance. Inserting these into our freezeout condition Eq. (20), we have

$$\frac{T_{\text{eq}}^2 m_\chi^4}{x_F^6} \cdot \frac{\alpha_{\text{eff}}^3}{m_\chi^5} \sim H_F \sim \frac{T_F^2}{M_{\text{Pl}}} \,. \tag{23}$$

Solving for the DM mass, assuming again $x_F \sim 20$, we find

$$\boxed{m_\chi \sim \alpha_{\text{eff}} \left( T_{\text{eq}}^2 M_{\text{Pl}} \right)^{1/3} \sim \alpha_{\text{eff}} \cdot (100 \, \text{MeV}).} \tag{24}$$

The mass-coupling relationship is then very different from the previous case, involving a generalized geometric mean between the two unrelated scales of $T_{\text{eq}}$ and $M_{\text{Pl}}$. Now it happens to be the case that if the self-coupling of DM is $\alpha_{\text{eff}} \sim \mathcal{O}(1)$, of order the strong coupling, then the strong scale emerges. Instead of a Weakly Interacting Massive Particle, we have arrived at a Strongly Interacting Massive Particle. Said another way, instead of a WIMP, we now have a SIMP. In analogy to the WIMP miracle, this is then dubbed 'the SIMP miracle'.

This DM candidate is much lighter than is typical for WIMPs, typically in the MeV to GeV mass range, with very different interactions. The freezeout picture remains the same as in the case of the WIMP. At what temperature does the SIMP freezeout? Even this remains similar to the case of the WIMP—it is freezing out when it is NR, at temperatures a factor of $\sim 20$ beneath its mass. Let's see this. We will use the instantaneous freezeout approximation, where we take the number density of DM to be that of equilibrium at the time of freezeout. This is a good proxy for the process at hand, and will give us good intuition. Since the equilibrium number density of DM in the NR regime is exponentially suppressed as in Eq. (5), the freezeout condition Eq. (20) reads

$$n_\chi^2 \langle \sigma v^2 \rangle_{3 \rightarrow 2} \sim (m_\chi T)^3 e^{-2x_F} \frac{\alpha_{\text{eff}}^3}{m_\chi^5} \sim H_F \sim \frac{T_F^2}{M_{\text{Pl}}} \sim \frac{m_\chi^2}{x_F^2 M_{\text{Pl}}} \,. \tag{25}$$

All we have to know is that $x_F = m_\chi / T_F$ is sitting in the exponent, and so solving for it we find that just as is the case for the WIMP, it is logarithmic in the parameters, $x_F \sim \frac{1}{2} \text{Log}(\text{parameters})$, resulting in $x_F \sim 20$ over a broad range of parameters, as we have used above. Indeed this feature holds for many DM mechanisms following similar logic.

More generally, we can think of an $n \rightarrow 2$ process of $\chi$ self-interactions. Such a process would be relevant if for instance DM is a fermion or if there is a $Z_2$ symmetry prohibiting the

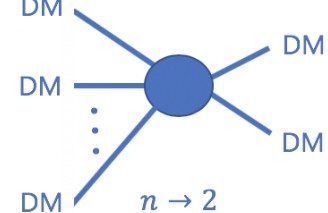

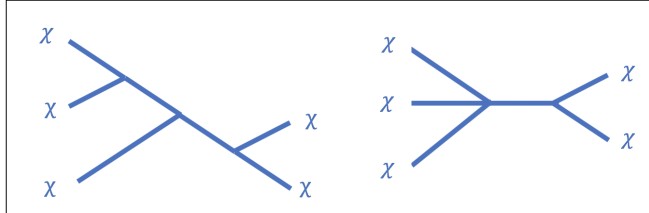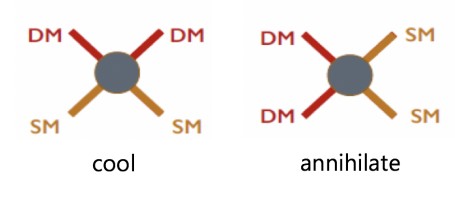

Figure 2: *Left:* Topologies for $3 \to 2$ interactions in the toy $Z_3$ scalar model. *Right:* Cooling versus annihilating.

$3 \to 2$ process from occurring. Consider the $n \to 2$ process as depicted here. We parameterize the thermally averaged cross section which enters the collision term by

$$\langle \sigma v^{n-1} \rangle_{n \to 2} \equiv \frac{\alpha^n}{m_\chi^{3n-4}} . \tag{26}$$

The dimensions can be understood from detailed balance as follows. In thermal equilibrium, detailed balance tells us that the rate for the forward process equals that of the backwards process. Considering the collision terms in the BE, this means that

$$\left( n_\chi^{\text{eq}} \right)^n \langle \sigma v^{n-1} \rangle_{n \to 2} - \left( n_\chi^{\text{eq}} \right)^2 \langle \sigma v \rangle_{2 \to n} = 0 \tag{27}$$

$$\Rightarrow \langle \sigma v^{n-1} \rangle_{n \to 2} = \left( n_\chi^{\text{eq}} \right)^{2-n} \langle \sigma v \rangle_{2 \to n} .$$

Now, $\langle \sigma v \rangle_{2 \to n}$ is an ordinary cross section with mass dimensions -2, and each power of number density carries mass dimension 3. Combined, we then find that the mass dimension of the $n \to 2$ thermally averaged object is $4 - 3n$, justifying our parametrization of the dimensions in Eq. (26).

The freezeout condition in this case is

$$\Gamma_{n \to 2} \sim n_\chi^{n-1} \langle \sigma v^{n-1} \rangle_{n \to 2} \sim H_F , \tag{28}$$

and using the tricks we developed previously leads to the mass-coupling relationship

$$\boxed{ m_\chi \sim \alpha_{\text{eff}} \left( T_{\text{eq}}^{n-1} M_{\text{Pl}} \right)^{1/n} , } \tag{29}$$

a truly generalized geometric mean, with each power of $n_\chi$ in the freezeout condition carrying one power of $T_{\text{eq}}$ as in Eq. (15). For $n = 3$, we have the case studied in Eq. (24). For $n = 4$, we have $m_\chi \sim \alpha_{\text{eff}} \times (100 \text{ keV})$.

Of course, $n \to 2$ processes are less familiar than $2 \to 2$ processes. To illustrate how easily the former can happen, we can construct a simple toy model for $3 \to 2$ annihilations. Later, we will see how $3 \to 2$ processes can arise generically in dark sectors. Consider a single scalar field $\chi$ that is charged under a $Z_3$ symmetry. The Lagrangian then contains renormalizable interaction terms of the form $\chi^3$ and $|\chi|^4$. From these, we can construct two topologies of 5-point interactions which yield $3 \to 2$ annihilations, using either three 3-point interactions, or one 3-point and one 4-point interaction, as depicted in the left panel of Fig. 2. Indeed, the use of three 3-point interactions was the inspiration for parameterizing the $3 \to 2$ cross section in Eq. (22) with a (coupling-strength)$^3$.

If you've kept up so far, great! But I have been cheating you. In the derivation of the mass-coupling relationship for the SIMP, we have implicitly assumed that there was one temperature

that described the entire system. However, if all that was occurring was simply the $3 \to 2$ self-annihilation of $\chi$, this process would pump heat into the dark sector, raising its temperature. (More on this soon.) The dark matter cannot be completely secluded; there must be a way to shed the heat, namely to dump entropy. This can be done via interacting with other light states or with the Standard Model (SM), for instance through elastic scattering with SM particles (see right panel of Fig. 2). We want the elastic scattering to be active during the time of $3 \to 2$ freezeout, but on the other hand, if we turn the diagram on its side (see right panel of Fig. 2), we arrive at the ordinary WIMP annihilation process, which we do not want to be active. In other words, we want to cool but not annihilate. Can this be done? Apriori it would seem hard, since the cross sections between the processes are typically of similar magnitude, $\langle \sigma v \rangle_{\text{cool}} \sim \langle \sigma v \rangle_{\text{ann}} = \langle \sigma v \rangle$. Fortunately, rates are also proportional to number densities. In the cooling case, a DM particle has to meet a SM particle in order for the process to proceed, while in the annihilation case, a DM particle has to meet another DM particle:

$$\Gamma_{\text{cool}} \sim n_{\text{SM}} \langle \sigma v \rangle, \qquad \Gamma_{\text{ann}} \sim n_\chi \langle \sigma v \rangle. \tag{30}$$

*If* the DM scatters off of abundant particles that are relativistic, such as electrons, photons or neutrinos, then the rate for annihilation is suppressed by the NR number density of DM, while the cooling rate is unsuppressed, and so

$$\frac{\Gamma_{\text{ann}}}{\Gamma_{\text{cool}}} \sim \frac{n_\chi}{n_{\text{SM}}} \sim e^{-m_\chi/T} \sim 10^{-8} \ll 1, \tag{31}$$

where in the last step we used $x_F \sim 20$ at the time of freezeout. As a result, we can easily cool but not annihilate by scattering off of light SM species.

We can now arrive at a set of conditions that must hold in order for the SIMP mechanism to work. On the one hand, we want cooling to be active at the time of $3 \to 2$ freezeout, and on the other had, we want the related $2 \to 2$ annihilations to be suppressed:

$$\left. \frac{\Gamma_{\text{cool}}}{\Gamma_{3 \to 2}} \right|_{T_F} \gtrsim 1, \qquad \left. \frac{\Gamma_{\text{ann}}}{\Gamma_{3 \to 2}} \right|_{T_F} \lesssim 1. \tag{32}$$

Parameterizing the SIMP-SM interactions by $\langle \sigma v \rangle \equiv \epsilon^2/m_\chi^2$, with $\epsilon$ indicating the size of the portal between the two sectors, we find a range of the interaction strength $\epsilon$ such that everything works, $\epsilon_{\text{min}} \lesssim \epsilon \lesssim \epsilon_{\text{max}}$, with

$$\begin{aligned}
\epsilon_{\text{min}} &\sim& \alpha_{\text{eff}}^{1/2} \left( \frac{T_{\text{eq}}}{M_{\text{Pl}}} \right)^{1/3}, \\
\epsilon_{\text{max}} &\sim& \alpha_{\text{eff}} \left( \frac{T_{\text{eq}}}{M_{\text{Pl}}} \right)^{1/6}.
\end{aligned} \tag{33}$$

The resulting range in the $\epsilon$-$m_\chi$ plane is depicted in the left panel of Fig. 3. When $\epsilon$ is too small, kinetic equilibrium is not maintained between the DM and the SM; when $\epsilon$ is too large, ordinary $2 \to 2$ annihilations take over; but for $\epsilon$ values in the broad available range, the precise value of $\epsilon$ plays no meaningful role, and SIMP dark matter emerges.

In summary, in the SIMP mechanism there are two important knobs. Strong $3 \to 2$ self-annihilations freeze out and control the DM relic abundance, while at the same time entropy-dumping processes such as elastic scattering off of light SM particles are active during the time of freezeout. The $3 \to 2$ process decouples first, while the elastic scattering process decouples second. What would happen if the order was reversed? To answer this, we will first delve into cannibals.

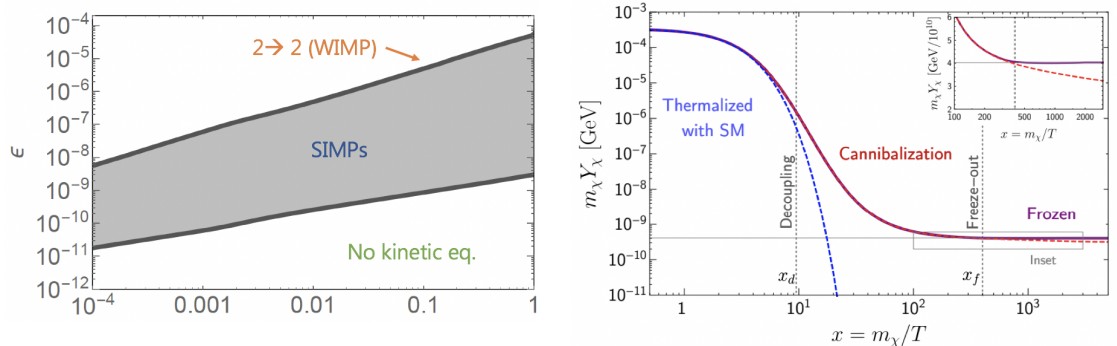

Figure 3: *Left:* Range of couplings between the DM and the SM where SIMP DM emerges. *Right:* Reproduced from [7], the DM yield as a function of SM temperature for ELDER DM, for DM mass 10 MeV, $\epsilon \sim 10^{-9}$ and strong $\alpha_{\text{eff}}$ (solid purple).

## 5 Cannibalism

Consider the $3 \to 2$ self-annihilations of DM; what if there were no light abundant species to dump entropy into? In this case, there is no reason for the DM to have the same temperature as the SM. We denote the temperature of the DM by $T_\chi$, to differentiate from $T$ the SM bath temperature. Once the DM becomes NR, $T_\chi < m_\chi$, the $3 \to 2$ process heats up the dark matter, raising its temperature $T_\chi$. By how much does it heat up? Since the $\chi$ particles do not speak with other particles, amongst themselves their comoving entropy is conserved, and so $s_\chi a^3 =$ constant. Now, we can write the entropy of the DM as

$$s_\chi = \frac{\rho_\chi + P_\chi}{T_\chi} \approx \frac{\rho_\chi}{T_\chi} = \frac{m_\chi n_\chi}{T_\chi} \sim \frac{m_\chi}{T_\chi} \left(m_\chi T_\chi\right)^{3/2} e^{-m_\chi/T_\chi}, \tag{34}$$

where we have used that the DM is in the NR regime and has vanishing chemical potential. Since $a \sim 1/T$, conservation of entropy thus implies

$$T_\chi \propto \frac{1}{\log a} \sim \frac{1}{\log(1/T)}. \tag{35}$$

Namely, the temperature of $\chi$ is growing exponentially compared to the temperature of the SM bath! This is dubbed 'cannibalism' – the $\chi$'s are eating themselves up to stay warm [9].

We can estimate the abundance of DM in this case. Denoting quantities today by '0', we can redshift to freezeout:

$$\frac{\rho_\chi^0}{s^0} = \frac{m_\chi n_\chi^0}{s^0} = \frac{m_\chi n_\chi^F}{s^F} = \frac{T_\chi s_\chi^F}{s^F}. \tag{36}$$

And the abundance of DM is then found to be (plugging in numbers and all factors)

$$\Omega_\chi = \frac{\rho_\chi^0}{\rho_c} = \frac{T_\chi s_\chi^F}{\rho_c} \frac{s^0}{s^F} \simeq 0.6 \frac{m_\chi/\text{eV}}{x_{\chi_F}} \left(\frac{s_\chi^F}{s^F}\right), \tag{37}$$

with $\rho_c$ the critical density and $x_\chi = m_\chi/T_\chi$. We learn that this scenario predicts very light DM unless there is a large entropy ratio between the dark and visible sectors (*e.g.* through large temperature differences or large ratio between the number of dofs in each).

# 6 From Cannibals to ELDERs

We just saw that a secluded bath of particles $\chi$ undergoing $\chi\chi\chi \to \chi\chi$ self-annihilations experiences cannibalisation. Suppose at some time before the $3 \to 2$ process shuts off, the dark bath was in thermal equilibrium with the SM, and the two sectors decoupled at some temperature $T_d = T_{\chi_d}$. Since the two sectors redshift the same, one has $s_{\chi_F}/s_F = s_{\chi_d}/s_d$, and so Eq. (37) can be written in terms of the entropy ratios at the time the sectors decoupled,

$$\Omega_\chi \simeq 0.6 \frac{m_\chi/\text{eV}}{x_{\chi_F}} \left( \frac{s_\chi^d}{s^d} \right). \tag{38}$$

This entropy ratio scales as

$$\frac{s_\chi^d}{s^d} \sim \begin{cases} \mathcal{O}(1) \quad (\text{more accurately, ratio of dof}), & \text{if } T_d \gg m_\chi \quad (\text{R}), \\ \left( \frac{m_\chi}{T_d} \right)^{5/2} e^{-m_\chi/T_d}, & \text{if } T_d \ll m_\chi \quad (\text{NR}). \end{cases} \tag{39}$$

When do the two sectors decouple? Roughly speaking, decoupling occurs when elastic scattering between the sectors, such as $\chi\gamma \to \chi\gamma$, is of order Hubble. We can then estimate the decoupling temperature by

$$\Gamma_{\text{el}} \sim n_\gamma \langle \sigma v \rangle_{\text{el}} \sim H_d \sim \frac{T_d^2}{M_{\text{Pl}}}. \tag{40}$$

Solving for the elastic scattering cross section, we find it is inversely proportional to the decoupling temperature,

$$\langle \sigma v \rangle_{\text{el}} \sim \frac{1}{T_d M_{\text{Pl}}} \sim \frac{m_\chi}{T_d} \left( \frac{1}{m_\chi M_{\text{Pl}}} \right). \tag{41}$$

Putting all the pieces together, for DM that decouples when non-relativistic, we arrive at the DM relic abundance

$$\Omega_\chi \propto e^{-\langle \sigma v \rangle_{\text{el}} \cdot \text{stuff}}, \tag{42}$$

namely a candidate where the abundance of DM is exponentially dependent on the elastic scattering cross section. This candidate is dubbed an 'ELastically DEcoupling Relic', or in short, an ELDER [7, 10].

In ELDER DM, much as in the SIMP mechanism, there are two knobs: $3 \to 2$ self-annihilations of DM, and $2 \to 2$ elastic scattering off of SM particles. For a SIMP, the former decouples first, and the latter decouples second, while for an ELDER, the latter decouples first, and the former decouples second. The evolution of the number density for ELDER DM is shown in the right panel of Fig. 3. At early times, the elastic scattering between the SM and the dark matter is active, and both sectors share a common temperature. The sectors decouple when elastic scattering stops, at which point the DM is cannibalizing away through the $3 \to 2$ annihilations. At a later point, the $3 \to 2$ process freezes out, and we are left with a constant relic abundance.

We can establish a phase diagram depending on the relative strength of the self-interactions compared to the DM-SM interactions, as shown in Fig. 4. As we change the relative size of these couplings, the theory flows from a regime where it is WIMP-like to a regime where it is SIMP-like, to a regime where it is an ELDER. We learn that these different mechanisms for DM candidates are simply different phases in the phase-space of interactions.

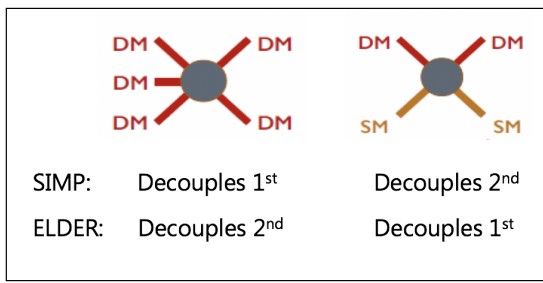

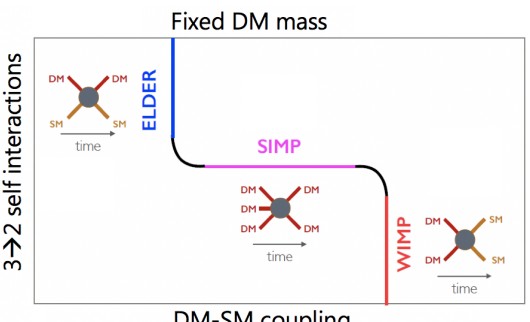

Figure 4: *Left:* SIMPs versus ELDERs. *Right:* The phase diagram of DM mechanisms in the plane of DM-SM couplings versus $3 \rightarrow 2$ self-couplings of DM, at fixed DM mass.

## 7 Dark Sectors

Thus far, we have developed several *mechansims* for DM freezeout and understood their behavior in terms of mass and couplings. We now move to discuss *models*—theories that realize these mechanisms. [Supersymmetry is the poster child for WIMPs, and we already saw a toy $Z_3$ model for SIMPs (see left panel of Fig. 2).] As we will see, the mechanisms we described are *generic* in theoryland. Consider the SM that we all know and love. It is a whole zoo of particles, governed by a beautiful symmetry structure of gauge symmetries, $SU(3)_c \times SU(2)_W \times U(1)_Y$.

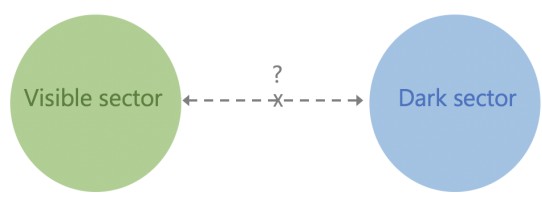

Why couldn't the dark sector similarly have many new particles, perhaps governed similarly by its own gauge symmetries? Inspired by the SM, maybe the dark sector has a dark version of QCD, $SU(3)_D$. In fact, it doesn't even have to be so QCD-like, it could be any $SU(N_c)$, $SO(N_c)$ or $Sp(N_c)$ dark gauge symmetry. Perhaps there is also a dark version of QED, a dark $U(1)_D$? This would give a dark photon $V$ that could kinetically mix with the ordinary photon, yielding a portal of interaction between the dark and SM sectors. The theories we will be talking about are thus strongly coupled gauge theories, which I will collectively call QCD-like theories, or in short, 'dark QCD'.

A minimal and simple dark sector could be just a dark $U(1)_D$—a dark version of QED, with dark particles charged under it. There would be kinetic mixing, $\mathcal{L} \supset -\frac{\epsilon}{2} F_{\mu\nu} F'^{\mu\nu}$ (with prime indicating the dark photon field strength) and a dark gauge coupling $e_D$ (or dark coupling strength $\alpha_D$), which would allow the SM and the DM to communicate through ex-

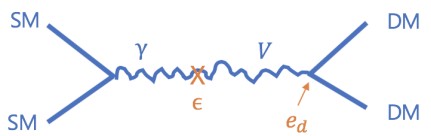

change of photon-dark photon. There could be non-abelian symmetries too; QCD-like dark sectors. These are rich theories, and so present a rich playground for many DM mechanisms and processes to occur.

QCD-like theories have many mesons, similarly to the SM ($\pi, K, \rho...$). The Pseudo Nambu Goldstone Bosons (PNGBs) of these theories, which we will collectively call dark pions, can play the role of DM. Consider for instance $2 \rightarrow 2$ processes that can occur in these theories, as shown in Fig. 5. Forbidden channels, WIMP-like annihilations, elastic scattering and semi-annihilations are all possible. In each case, one can compute from the Lagrangian of the theory (written in terms of the parameters of the model—$\alpha_d$, $m_\pi$, $f_\pi$, $m_V$, etc.) the cross section needed to explain the relic abundance of DM, *e.g.* $\langle \sigma v \rangle = f(\text{parameters})$. Then one

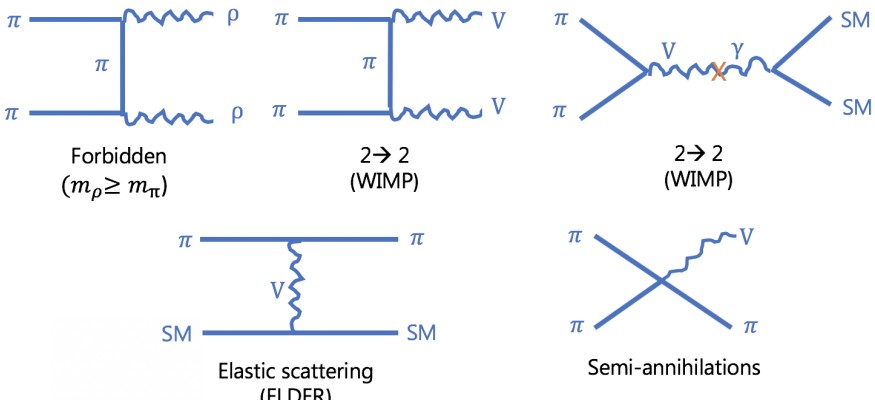

Figure 5: Examples of $2 \rightarrow 2$ processes that can occur in QCD-like dark sectors, with $\pi, \rho$ indicating dark mesons, and the dark pions are the DM.

can translate the cross section we previously developed at the mechanism level into the the theory at hand, to understand what parameters are needed.

For instance, consider the $3 \rightarrow 2$ process of the SIMP. While such a process might seem the most 'exotic' as it is less familiar than $2 \rightarrow 2$ processes, such interactions do occur in the SM itself! To see this, here is a quick reminder of QCD. QCD is an $SU(3)$ gauge theory with 3 light flavors, $u, d, s$. The theory has an approximate $SU(3)_L \times SU(3)_R$ global symmetry. At low energy, the theory confines and chiral symmetry breaking occurs, breaking the global symmetry down to its diagonal component: $SU(3)_L \times SU(3)_R \rightarrow SU(3)_{\text{diag}}$. (Of course, in the SM these global symmetries are only approximate because the quark masses are not degenerate.) As a result, there are 8 PNGBs corresponding to the 8 broken generators – these are the kaons, the pions and the eta. This theory has 5-point interactions: two kaons can annihilate into 3 pions, $K^+ K^- \rightarrow \pi^+ \pi^- \pi^0$. This occurs through a topological term in the Lagrangian called the Wess-Zumino-Witten (WZW) term [11–13]. If you calculate the rate, you find its just right to be a SIMP if the mass of the particles is of order $\sim 100$ MeV!

Inspired by this, let's examine QCD-like theories. Consider for instance an $SU(N_c)$ gauge theory with $N_f$ flavors with degenerate mass. (This degenerate mass for the dark quarks will ensure the residual symmetry is exact and will thus stabilize the DM.) The theory has an (exact) $SU(N_f)_R \times SU(N_f)_L$ global symmetry, which after chiral symmetry breaking occurs breaks to $SU(N_f)_{\text{diag}}$ which is now exact. There are $N_\pi$ PNGBs from the broken generators — these are the dark pions, which can be the DM. The theory has 5-point interactions through the WZW term, which take a particular form:

$$\mathcal{L}_{\text{WZW}} = \frac{N_c}{15\pi^2 f_\pi^5} \epsilon^{\mu\nu\rho\sigma} \text{Tr}\left[\pi \partial_\mu \pi \partial_\nu \pi \partial_\rho \pi \partial_\sigma \pi\right], \tag{43}$$

with $\pi = \pi^a T^a$, with $T^a$ the broken generators and $a = 1 \dots N_\pi$. The coefficient of this 5-point interaction term is written in terms of the pion decay constant. One can similarly do this for the other gauge groups $SO(N_c)$ and $Sp(N_c)$. In all cases, provided that there are enough flavors ($N_f \geq 3$ for $SU(N_c)$ and $SO(N_c)$, and $N_f \geq 2$ for $Sp(N_c)$), the topological condition is met and the WZW term exists as in Eq. (43), giving 5-pion interactions. We have thus found that $3 \rightarrow 2$ processes are generic in QCD-like theories [14]. Since in a sense, these dark pion constructions are the simplest version in generic theories for SIMP DM, these scenarios are dubbed 'the SIMPlest miracle'.

We can compute the thermally averaged cross section for the $3 \rightarrow 2$ process in these theories, written now in terms of the parameters of the theory—the mass $m_\pi$ and decay constant

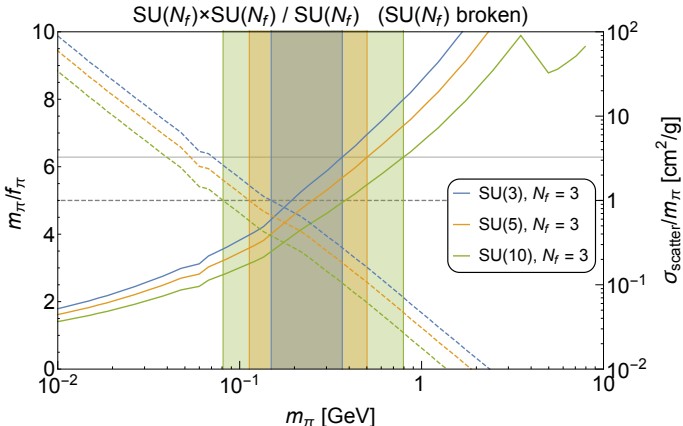

Figure 6: Reproduced from [14]. **Solid curves:** the solution to the Boltzmann equation of the $3 \to 2$ system, yielding the measured dark matter relic abundance for the pions, $m_\pi/f_\pi$ as a function of the pion mass (left axis). **Dashed curves:** the self-scattering cross section along the solution to the Boltzmann equation, $\sigma_{\text{scatter}}/m_\pi$ as a function of pion mass (right axis). All curves are for selected values of $N_c$ and $N_f$, shown here for an $SU(N_c)$ gauge group with broken $SU(N_f)$ flavor symmetry. The solid horizontal line depicts the perturbative limit of $m_\pi/f_\pi \lesssim 2\pi$, providing a rough upper limit on the pion mass; the dashed horizontal line depicts the bullet-cluster and halo shape constraints on the self-scattering cross section, placing a lower limit on the pion mass. Each shaded region depicts the resulting approximate range for $m_\pi$ for the corresponding symmetry structure.

$f_\pi$ of the pions—and find:

$$\langle \sigma v^2 \rangle_{3 \to 2} = \frac{5\sqrt{5}}{2\pi^5} \frac{N_c^2}{x_F^2} \frac{m_\pi^5}{f_\pi^{10}} \left( \frac{t^2}{N_\pi^3} \right), \tag{44}$$

where the last factor is a combinatorical factor that depends on the gauge group and on $N_f$ (see [14]). In other words, we are able to translate the mechanism-level parameterization of $\alpha_{\text{eff}}$ in Eq. (22) into the parameters of the theory, with $\alpha_{\text{eff}} \sim \#(m_\pi/f_\pi)^{10/3}$ in this case. Producing the correct relic abundance of DM translates to the solid curves in the $m_\pi/f_\pi - m_\pi$ plane shown in Fig. 6. Since perturbativity breaks down around $m_\pi/f_\pi \sim 2\pi$ (solid horizontal curve), the maximal dark pion mass is of order $\sim$GeV before non-perturbative effects contribute.

What about self-scattering? One would expect that in strongly coupled theories with strong $3 \to 2$ self-annihilations, there will be unsuppressed contributions to $2 \to 2$ self-scattering processes as well. Self-scattering amongst DM particles can distort dynamics in dark matter halos, and typically limits

$$\frac{\sigma_{\text{scatter}}}{m_\chi} \lesssim 1 \frac{\text{cm}^2}{\text{g}}. \tag{45}$$

There are also long-standing puzzles in structure formation such as core-vs-cusp and the too-big-to-fail puzzles. While it is possible that these are addressed by unaccounted-for baryonic effects, it is also possible to accommodate these by self-interactions of DM of order the limit in Eq. (45), which corresponds to roughly $\sim$barn/MeV—namely interactions of order the strong force interactions. (See [15] for a nice review on this subject.)

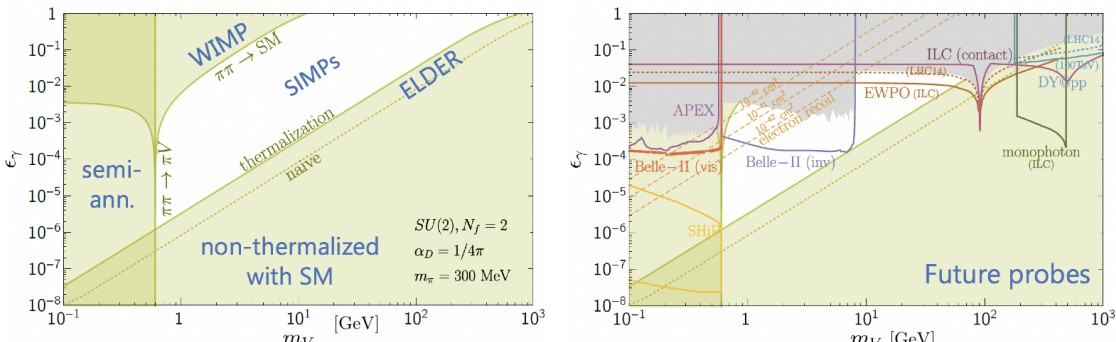

Figure 7: Reproduced from [16]. *Left:* Dark photon parameter space for fixed gauge group, dark coupling and dark pion mass, showing how different regions realize different DM mechanisms. *Right:* Constraints on the parameter space (shaded gray) and future probes (solid colored curves).

In the context of the QCD-like theories we have been discussing, self-scattering is thus an important constraint on the parameter space, and possibly an unavoidable signal as well. The Lagrangian of these theories after chiral symmetry breaking contains several pieces that give rise to 4-point pion interactions, and thus to self-scatterings:

$$\mathcal{L} \supset \frac{m_\pi^2}{12 f_\pi^2} \mathrm{Tr}\pi^4 - \frac{1}{6 f_\pi^2} \mathrm{Tr}\left(\pi^2 \partial_\mu \pi \partial^\mu \pi - \pi \partial_\mu \pi \pi \partial^\mu \pi\right). \tag{46}$$

From this, one finds

$$\sigma_{\text{scatter}} = \frac{m_\pi^2}{32\pi^2 f_\pi^4}\left(\frac{a^2}{N_\pi^2}\right), \tag{47}$$

with the last factor again a combinatorical one that depends on the gauge group and the number of flavors $N_f$. Since for a given DM mass, $m_\pi/f_\pi$ is dictated by the relic abundance, the self-scattering cross-section is then determined as well. The dashed curves in Fig. 6 show the self-scattering cross-section along the relic abundance solution with the corresponding constraint, yielding a lower limit on the dark pion mass. Combined, we learn that SIMP dark pions in QCD-like theories point to the strongly interacting regime of the theory, with DM masses in the MeV-GeV mass range, typically of $\mathcal{O}$(few 100 MeV).

One can also use other degrees of freedom in QCD-like theories, such as glueballs, to study $3 \to 2$ processes, see *e.g.* [9,17,18]. Regarding the interaction portal between the dark pions and the SM, as neccessary for the SIMP mechanism, the dark photon and axion portals have been studied in [16] and [19], respectively.

The QCD-like theories we described not only give us generic realizations of various DM mechanisms—they are predictive as well. Consider for instance QCD-like theories with a dark $U(1)_D$ as well. (Indeed, one can embed this gauged $U(1)_D$ into the residual flavor symmetry of the QCD-like theories, see [16] for further details.) The dark photon $V$ can kinetically mix with the ordinary photon, giving interactions between the sectors. Fig. 7 shows for a fixed gauge group and dark pion mass, how different regions of this very simple setup realize various types of DM mechanims, be it WIMPs, SIMPs, ELDERs, semi-annihilations and more. The parameter space is constrained by many experiments and importantly can be probed by many other planned future experiments, from high-energy collider to low-energy colliders, through fixed-target and beam dump experiments, as well as direct detection.

One can go even further than this: we can hope to perform *spectroscopy* of the dark sector states themselves. To see this, we again draw an analogy from QCD. How do we know the

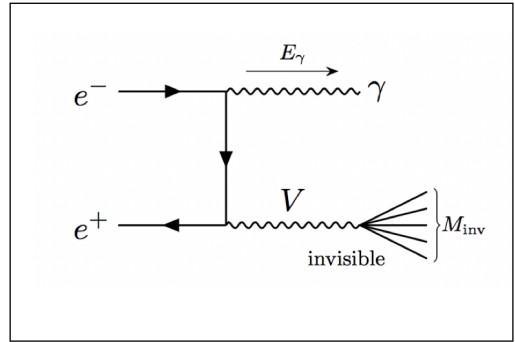
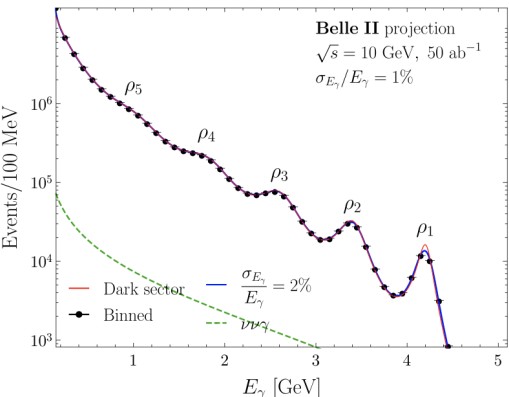

Figure 8: Reproduced from [20]. *Left:* Mono-photon production at a lepton collider. *Right:* The cross section for $e^+e^- \to \gamma + \text{inv}$ at $\sqrt{s} = 10$ GeV for an $SU(2)_d \times U(1)_d$ gauge theory with 4 Weyl fermions, dark photon mass $m_V = 12$ GeV, DM mass $m_\pi = 1$ GeV, lightest dark $\rho$ meson $m_{\rho_1} = 4$ GeV, kinetic mixing $\epsilon = 10^{-2}$ and dark coupling $\alpha_D = 0.1$, with 1% energy resolution. The SM background $e^+e^- \to \gamma \nu \nu$ is shown in green.

beautiful resonance structure of QCD? We smash $e^+e^-$ on each other, which goes to lots of 'stuff'. By scanning the center of mass energy of the collision, we scan the resulting 'stuff' of the final products. You could ask yourself: how could we see this at a fixed energy machine? The answer is: tack on a photon! The photon in the final state is a tracer for the system it recoils against. For that, one doesn't have to be able to see the system it recoils against—it could be dark 'stuff' too, as in the left panel of Fig. 8. Mono-photon events at fixed energy machines thus allow us to measure the spectrum of the invisible system it recoils against,

$$E_\gamma = \frac{\sqrt{s}}{2}\left(1 - \frac{M_{\text{inv}}^2}{s}\right). \tag{48}$$

Performing spectroscopy of dark sectors via mono-photon searches would be possible at low-energy colliders, such as the Belle-II experiment at KEK. An example of a SIMP-like spectrum that could be observed at Belle-II is shown in the right panel of Fig. 8, demonstrating the power of this approach.

# 8 Conclusion

Focusing on high-point interactions, we have developed SIMP, cannibal and ELDER dark matter, and shown that these novel dark matter candidates are generic and predictive in theoryland. The emphasis has been on giving you the tool kit to quickly estimate and understand the expected behavior and characteristics of new dark matter ideas. Most of all, I hope I have provided you with tools for when you come up with the next amazing dark matter candidate, which I look forward to learning about.

## Acknowledgements

I thank the organisers of the Les Houches Summer School 2021 for inviting me to teach, and the students at the school, who made it so much fun. It was a true pleasure. I thank Mrunal Prashant Korwar for useful comments on the manuscript. I thank the Israel Science

Foundation (grant No. 1112/17), the Binational Science Foundation (grant No. 2016155), the I-CORE Program of the Planning Budgeting Committee (grant No. 1937/12), and the Azrieli Foundation for their support.

## A  Crocodile Challenge

If you ever met me, you know I love crocodiles (more accurately, all things crocodilian). At the beginning of this school I posed the crocodile challenge: to have a reason for a crocodile Feynamnn diagram to appear in a physics paper. The closest I have ever been to succeeding in this challenge is shown in Fig. 9, which came up while working on the axion portal for SIMPs. Unfortunately in the particular case we studied the diagram vanished and so did not make it to the final paper draft. I am grateful to Eric Kuflik who discovered this diagram, and to the Les Houches Summer School Lecture Notes Series who enabled this crocodile to be seen in the (e)printed world.

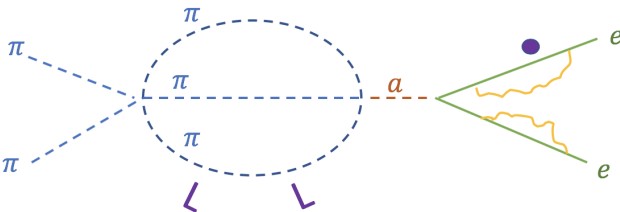

Figure 9:  The crocodile challenge.

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
