# Peer review of "SIMP Dark Matter"

_SciPost Physics Lecture Notes, doi:SciPost Phys. Lect. Notes 59 (2022)_

## Round 1 · Referee Report · Anonymous (Referee 1) · 2022-4-5

Report

This is an exceptionally well-written set of lecture notes, with the main points clearly and elegantly explained with an appropriate dose of humor. The level is entirely appropriate for graduate students, and this set of notes will be an excellent addition to the literature.

---

## Round 1 · Referee Report · Hyun Min Lee (Referee 2) · 2022-4-6

Report

The paper is well written and it would provide a good reference for starting a research on dark matter production in early cosmology. Thus, I recommend a publication of the paper.

---

## Round 1 · Referee Report · Anonymous (Referee 3) · 2022-4-8

Strengths

These notes are extremely pedagogical and provide a warm welcome into this topic. I will probably refer my own students to these notes in the future.

Report

The notes cover a relevant topic of interest to the community and provide cohesive explanations of the relevant phenomena. Therefore, the acceptance criteria are clearly met and I recommend that these notes proceed through the acceptance process.

Requested changes

My only minor constructive comments are:

1) Since these are notes intended for students, it may be useful to expand the list of references slightly just so that students have links to click and go down the rabbithole if they are confused or want to learn more about a particular statement made in the text or appearing in a plot. I found the current list of references to be somewhat sparse, but that is just my opinion.

2) The only part of the notes that I found a bit unclear or where there could be some expanded commentary was in the cannibals section, where the entropy is computed assuming the cannibals are nonrelativistic. Of course, if things are heating up exponentially, at some point this assumption will cease to be true, and I think this would merit addressing (briefly).

---

## Editorial Decision

published